# Exploring Risk Factors of Recall-Associated Foodborne Disease Outbreaks in the United States, 2009–2019

**DOI:** 10.3390/ijerph19094947

**Published:** 2022-04-19

**Authors:** Emily Sanchez, Ryan B. Simpson, Yutong Zhang, Lauren E. Sallade, Elena N. Naumova

**Affiliations:** 1Friedman School of Nutrition Science and Policy, Tufts University, Boston, MA 02111, USA; ryan.simpson@tufts.edu (R.B.S.); yutong.zhang@tufts.edu (Y.Z.); lauren.sallade@tufts.edu (L.E.S.); 2Army Medical Department Student Detachment, U.S. Army Medical Center of Excellence, Fort Sam Houston, San Antonio, TX 78234, USA

**Keywords:** data credibility, foodborne outbreaks, food recalls, National Outbreak Reporting System (NORS)

## Abstract

Earlier identification and removal of contaminated food products is crucial in reducing economic burdens of foodborne outbreaks. Recalls are a safety measure that is deployed to prevent foodborne illnesses. However, few studies have examined temporal trends in recalls or compared risk factors between non-recall and recall outbreaks in the United States, due to disparate and often incomplete surveillance records in publicly reported data. We demonstrated the usability of the electronic Foodborne Outbreak Reporting System (eFORS) and National Outbreak Reporting System (NORS) for describing temporal trends and outbreak risk factors of food recalls in 1998–2019. We examined monthly trends between surveillance systems by using segmented time-series analyses. We compared the risk factors (e.g., multistate outbreak, contamination supply chain stage, pathogen etiology, and food products) of recalls and non-recalls by using logistic regression models. Out of 22,972 outbreaks, 305 (1.3%) resulted in recalls and 9378 (41%) had missing recall information. However, outbreaks with missing recall information decreased at an accelerating rate of ~25%/month in 2004–2009 and at a decelerating rate of ~13%/month after the transition from eFORS to NORS in 2009–2019. Irrespective of the contaminant etiology, multistate outbreaks according to the residence of ill persons had odds 11.00–13.50 times (7.00, 21.60) that of single-state outbreaks resulting in a recall (*p* < 0.001) when controlling for all risk factors. Electronic reporting has improved the availability of food recall data, yet retrospective investigations of historical records are needed. The investigation of recalls enhances public health professionals’ understanding of their annual financial burden and improves outbreak prediction analytics to reduce the likelihood and severity of recalls.

## 1. Introduction

Every year, ~48 million Americans get sick, 128,000 are hospitalized, and 3000 die from foodborne diseases [1]. From 2013 to 2018, the US Department of Agriculture’s (USDA) Economic Research Service estimated that the value of preventing foodborne illnesses, a measure of demand for reduction in mortality risk, increased by 12%, from $12.8 to $14.4 billion (USD), respectively [1]. These estimates account for inpatient and outpatient hospital costs and costs of prescription drugs and medical supplies used to treat infected persons [1].

One safety measure deployed to prevent foodborne illnesses is food recalls, or when a manufacturer or distributor voluntarily removes food products from commerce due to their expected risk to human health [2]. In 2011, the Food Marketing Institute and Grocery Manufacturers Association reported that food recalls cost ~$10 M/recall in direct costs to food companies [3]. The study further noted that ~23% of annual recalls exceed ~$30 M/recall in direct costs, which accounts for product retrieval, storage destruction, and regulatory notifications throughout the supply chain [3,4]. However, these costs underestimate recalls’ total financial burden, as direct costs exclude government fines, food-safety hygiene compliance penalties, lawsuits, lost sales, and damaged brand reputation [5,6,7]. Indirect costs have long-lasting impacts; for example, a 2010 Harris Interactive Poll found that 55% of consumers would switch brands if consuming a recalled product and 36% of consumers would never purchase the product or brand again if they learned of a recalled product [3]. 

The early detection of outbreaks and identification of contaminated food products is crucial in reducing the health burdens of foodborne outbreaks. While research has explored temporal trends and risk factors associated with foodborne outbreaks, less is known regarding food recalls [8,9,10]. The USDA’s Food Safety and Inspection Service (FSIS) and Department of Health and Human Services’ Food and Drug Administration (FDA) are charged with monitoring and investigating recalls [11]. The FSIS routinely inspects and regulates meat, poultry, and processed egg recalls, while the FDA regulates all other products [12]. When conducting traceback investigations, both agencies publicly report the product manufacturer and ingredients of contaminated food products [12,13]. The FSIS also reports annual tallies of recalls by class (i.e., public risk level), suspected contaminant (i.e., pathogen, allergen, substance, or chemical/toxin), product type (i.e., beef, pork, poultry, etc.), and volume of food recalled [13]. The FDA provides event-based descriptive summaries of implicated food company name, a product description, product type, and recall termination date [12]. 

While the current literature on recalls largely explores their impact on consumer trust and behaviors, few publications provide in-depth descriptions of recall risk factors or any information on non-recall outbreaks [14,15]. In one study exploring the direct contribution of meat and poultry recalls to the food waste stream, the FSIS meat and poultry recalls from 1994 to 2015 were found to be predominantly attributed to *Listeria*, undeclared allergens, and Shiga toxin-producing *Escherichia coli* (STEC) [16]. Additional reports summarize outbreak investigations with brief mention of the outbreak resulting in a food recall or not [17,18,19]. Furthermore, resource allocation to investigate and report foodborne outbreaks may be determined by a public health agency’s mission and subsequently effect the frequency of outbreak-related recalls reported. That is, since the FSIS regulates meat, poultry and processed eggs, microbial sampling and testing programs are more routinely conducted for *Salmonella*, *Campylobacter*, *Listeria monocytogenes*, and STEC [20]. Lastly, despite analyzing implicated product types and ingredients, neither the USDA nor the FDA reports the supply chain stage of contaminant introduction or locations of food preparation and consumption by ill persons. These data are critical for improving outbreak prediction analytics to enhance food traceability and expedite public health responses to foodborne outbreaks and recalls in accordance with the FDA’s 2020 New Era of Smarter Food Safety Blueprint [21,22]. Comparisons between the risk factors of recall and non-recall outbreaks can improve food safety technology and interagency collaborative efforts for rapid outbreak response detection and mitigation [22]. 

In 2012/2013, the Government Accountability Office (GAO) reported that the USDA’s and the FDA’s recall surveillance systems were unreliably, inconsistently, and incompletely reporting recall record data, preventing its usage for analytical purposes [23]. For example, annual tallies and descriptive reports lack the comprehensiveness to extensively study recall record temporal trends and risk factors. Though agencies report recalls’ dates, reporting formats are not conducive to performing time-series analyses. This impedes data users from investigating the longitudinal impact of food safety policies and justifying strategies for continued regulatory oversight and enforcement [24]. Additionally, the inability to conduct time-series analyses prevents the inspection of recall seasonality imperative for public health professionals to prepare for and mitigate the intensity of seasonal foodborne outbreaks and illnesses [8,25,26,27,28]. While data-quality-related issues have improved per a 2018 GAO report, USDA agency officials still noted that data are poorly utilized within analytical workflows to inform food safety and consumption guidelines [29].

In fact, the most recent 2020/2021 GAO report emphasized that gaps in the USDA/FDA foodborne illness and recall surveillance stress the need for improved surveillance capacity on these topics by the Centers for Disease Control and Prevention (CDC) [30]. Though not responsible for investigating recalls, the CDC has conducted thorough event-based surveillance of waterborne and foodborne outbreaks since 1971 and 1973, respectively [31]. In 1998, CDC surveillance transitioned from the paper-based Foodborne Outbreak Reporting System (pFORS) to electronic reporting (eFORS) [32]. In contrast to the FSIS and FDA, the CDC’s records include data on where, when, how many persons, what food sources, and which pathogens are associated with outbreaks [32]. In November 2004, the CDC began identifying if outbreaks resulted in recalls and reporting recall-related traceback information [33]. In January 2009, the CDC integrated all outbreak surveillance data streams into the National Outbreak Report System (NORS), which monitors, tracks, and reports on 45 person-to-person, zoonotic, environmental, and unknown/indeterminate sources of outbreaks [32]. 

Despite the NORS’s comprehensiveness, public health professionals have only recently begun exploring the usability of NORS for describing recall records [34]. These studies have largely explored the likelihood of increased morbidity and mortality of recalls compared to non-recalls [34]. To the best of our knowledge, no studies have utilized recall record data to examine temporal trends, particularly before and after November 2004 or January 2009. Additionally, no studies have compared risk factors between recalls and non-recalls, perhaps due to the limited completeness of surveillance records [35]. Thus, further investigation of recall records reported by the CDC is warranted.

In this study, we demonstrated the use of recall record data from electronic Foodborne Outbreak Reporting System and National Outbreak Reporting System for investigating temporal trends and risk factors of food recalls in 1998–2019. First, we described monthly trends and seasonality of recalls, non-recalls, and outbreaks with missing recall information, using segmented negative binomial regression models with respect to three delineated periods: prior to November 2004 (surveillance reporting begins under eFORS), prior to January 2009 (foodborne outbreak reporting revised to include if any product was recalled from an outbreak), and after January 2009 (surveillance reporting begins under NORS). Next, we examined risk factors (e.g., multistate exposure outbreak, contamination supply chain stage, pathogen etiology, and food products) associated with recalls, using logistic regression models. Our findings highlight improvements in CDC surveillance reporting over time; however, they still note extensive incomplete records on food recalls.

## 2. Materials and Methods

### 2.1. Data Source

On 4 March 2021, we requested CDC surveillance records for foodborne and waterborne outbreaks from 1 January 1998 to 31 December 2019. The NORS Foodborne and Animal Contact Team investigated the accuracy and quality of data prior to distribution. Due to the volume of surveillance records requested, data were separated into 65 data tables, each corresponding to characteristics of the time, location, pathogen, preparation/consumption location, food ingredients, and food inspection methods of an outbreak. NORS used unique outbreak identifiers to harmonize records across tables and provided a comprehensive dictionary to describe variables and their units of measurement per table [36]. 

We created etiology-, state-, and county-specific binary indicator variables (1 = present, and 0 = absent). Etiology-specific variables included 29 pathogens (adenovirus, *Anisakis*, astrovirus, *Bacillus*, *Brucella*, *Campylobacter*, *Ciguatoxin*, *Clostridium*, *Cryptosporidium*, *Cyclospora*, *Entamoeba*, *Enterobacter*, *Enterococcus*, *E. coli*, *Giardia*, *Hepatitis*, *Listeria*, norovirus, *Proteus*, rotavirus, *Salmonella*, sapovirus, *Shigella*, *Staphylococcus*, *Streptococcus*, *Toxoplasma*, *Trichinella*, *Vibrio*, and *Yersinia*) and 5 toxins/poisons (shellfish poison, ciguatoxin, plant/herbal toxins, puffer fish tetrodotoxin, and scombroid toxin). States included the 50 US states; Washington, D.C.; and three US territories (Guam, Puerto Rico, and Republic of Palau). We created multi-etiology, multistate exposure, and multi-county variables for outbreaks associated with numerous etiologies or where an outbreak was caused by exposures in multiple states, respectively [37].

Reported recall record data included the type of food product, a description of the recalled product, and the product’s brand or lot number, a distinct combination of letters, numbers, or symbols that correspond to the complete history of the manufacturer, processing, packing, holding, and distribution of a product [38]. We examined the recall status according to three categories: outbreaks resulting in recalls (recalls), outbreaks not resulting in recalls (non-recalls), and outbreaks with missing recall information (missing).

Under NORS, recall-related traceback information was incorporated into the recall record. NORS classified 3 supply chain contamination points, namely before preparation (i.e., production, harvesting, packaging, and transporting), preparation (i.e., cooking, retail, consumption), and unknown. The before preparation category was further disaggregated into 3 subcategories: pre-harvest (e.g., traceback to producer farms and fields), preprocessing (e.g., traceback to leaking produce cleaning and storage facility), and unknown preparation. 

NORS reports food products by using the Interagency Food Safety Analytics Collaboration (IFSAC) classification scheme. Developed in 2011 by the CDC, FSIS, and FDA, IFSAC is a 5-level food-categorization hierarchy that specifies 234 food categories [39]. Level 1 refers to the coarsest categorization of food groups (e.g., aquatic animals, land animals, plants, and other foods), while Level 5 disaggregates groups by processing, preparation, and consumption type (e.g., fermented, cured, salt-cured, etc.).

NORS reported 23 locations where ill persons prepared and consumed contaminated foods implicated in causing an outbreak, including restaurants (e.g., fast food, sit-down, and other), function halls (e.g., private home, banquet facility, and caterer), community gathering areas (e.g., daycare, school, prison/jail, religious location, camp, picnic, and fair), and other locations (e.g., grocery store, workplace, nursing home, assisted living facility, hospital, and home) [40]. These categories were not mutually exclusive; we created dichotomous variables for each location, as well as multi-location food preparation or consumption.

### 2.2. Investigating Temporal Trends

We conducted a segmented time-series regression analysis to investigate monthly count trends and seasonality in outbreaks and recalls over our 22-year study period. We divided our study period into 3 critical periods according to 2 critical points: November 2004, or when the CDC began food recall reporting; and January 2009, or when surveillance transitioned from eFORS to NORS (Table 1). 

To estimate the mean monthly counts of outbreaks, recalls, non-recalls, and outbreaks missing recall information for the entire study period, we applied a generalized linear model with a negative binomial distribution and logarithmic link function:$$\mathrm{Model}\;1:\;\ln\left[E\left(Y_{i}\right)\right]=\beta_{0}$$
$$\mathrm{Model}\;2:\;\ln\left[E\left(Y_{i}\right)\right]=\beta_{0}+\beta_{1}t*z_{i}$$
where *Y_i_* is the estimated mean of *i*-outcome (e.g., outbreaks, recalls, non-recalls, and missing); $\beta_{0}$ is the estimated mean for *i*-outcome for the study period of 264 months; *t* is the consecutive time, in months, ranging from 1 to 264 sequentially; and *z* is a binary indicator variable for *i*-critical period ranging from 1 to 3, where the outcome of interest occurred (*z* = 1). By exponentiating the model’s intercept, we calculated the estimated mean, exp$\left\{\beta_{0}\right\}$, and their 95% confidence interval estimates, exp$\left\{\beta_{0}\pm 1.96se\right\}$. 

The study period of 264 months divided into three critical periods was marked with knots, or critical points where *a* represents the start of critical period 2 at study month 82 and *b* represents the start of critical period 3 at study month 132. Using the selected periods, we developed a segmented negative binomial regression model to examine the temporal trends across the three critical periods for all outcomes:$$\mathrm{Model}\;3:\;\ln\left[E\left(Y_{t,i}\right)\right]=\beta_{0}+\beta_{1}t+\beta_{2}\left(t-a\right)+\beta_{3}\left(t-b\right)$$
$$\mathrm{Model}\;4:\;\ln\left[E\left(Y_{t,i}\right)\right]=\mathrm{Model}\;3+\beta_{4}t^{2}+\beta_{5}{\left(t-a\right)}^{2}+\beta_{6}{\left(t-b\right)}^{2}$$
$$\mathrm{Model}\;5:\;\ln\left[E\left(Y_{t,i}\right)\right]=\mathrm{Model}\;4+\beta_{7}\sin\left(2\pi \omega t\right)+\beta_{8}\cos\left(2\pi \omega t\right)+\beta_{9}\sin\left(2\pi \omega \left(t-a\right)\right)+\;\beta_{10}\cos\left(2\pi \omega \left(t-a\right)\right)+\beta_{11}\sin\left(2\pi \omega \left(t-b\right)\right)+\beta_{12}\cos\left(2\pi \omega \left(t-b\right)\right)$$
where *Y_t_*_,*i*_ represents the monthly counts of *i*-outcome (e.g., outbreaks, recalls, non-recalls, and missing) in *t*-month; *t* is the consecutive time in months, ranging from 1 to 264, sequentially; and *a* and *b* are the locations of the critical points at 82 and 132 months, respectively. Moreover, *t*, (t−a), and (t−b); and $t^{2}$, ${\left(t-a\right)}^{2}$, and $\left(t-b\right){\;}^{2}$ are the linear and quadratic trends of continuous time-series variables in months, respectively. In addition, sin(2πωt), sin(2πω(t−a), and  sin(2πω(t−b) ; and cos(2πωt), cos(2πω(t−a)), and cos(2πω(t−b)) are the sinusoidal and co-sinusoidal harmonic terms, respectively, with a frequency of ω=1/M, where M=12 represents the length of the annual cycle in months. 

We assessed the contribution of linear and quadratic trend terms in Model 4. The linear term indicated overall increases ($\beta_{1}t\;$> 0, $\beta_{2}\left(t-a\right)\;$> 0, $\beta_{3}\left(t-b\right)\;$> 0) or decreases ($\beta_{1}t$ < 0, $\beta_{2}\left(t-a\right)\;$< 0, $\beta_{3}\left(t-b\right)$ < 0), while the quadratic term indicated acceleration ($\beta_{1}t^{2}$ > 0, $\beta_{2}{\left(t-a\right)}^{2}\;$> 0, $\beta_{3}{\left(t-b\right)}^{2}\;$> 0) or deceleration ($\beta_{1}t^{2}\;$< 0, $\beta_{2}{\left(t-a\right)}^{2\;}$< 0, $\beta_{3}{\left(t-b\right)}^{2}\;$< 0) within each critical period. We calculated the trend contribution by multiplying each coefficient by the trend-associated time unit to recover the corresponding predicted rates:$$TC_{i,j,m,k}=\frac{\left|\beta_{m}{\left(t-t_{k}\right)}^{j}\right|}{\left|\beta_{1}t\right|+\left|\beta_{2}\left(t-a\right)\right|+\left|\beta_{3}\left(t-b\right)\right|+\left|\beta_{4}t^{2}\right|+\left|\beta_{5}{\left(t-a\right)}^{2}\right|+\left|\beta_{6}{\left(t-b\right)}^{2}\right|}$$
where *TC_i_*_,*j*,*m*,*k*_ is the contribution of the *i*-outcome for *j*-trend (*j* = 1 for linear term, *j* = 2 for quadratic term) in the $\beta_{m}$ coefficient, with *m* ranging from 1 to 6 for *k*-continuous time series variable (e.g., *a* and *b*; for summary of model coefficients and diagnostics, see Appendix A). The trend terms across all critical periods were summed to 1.00 per outcome regression model. We determined seasonality by the significance of either harmonic term in Model 5.

### 2.3. Assessing Risk Factors Associated with Recalls

Based on the trend analyses, we found that very few non-recalls occurred as monthly counts of outbreaks missing recall information increased in Period 1. However, while monthly counts of non-recalls began to rise, outbreaks missing recall information decreased in Period 2. To better understand the risk factors associated with an outbreak resulting in a recall, we chose to conduct risk-factor analyses amongst recalls and non-recalls aggregated with outbreaks missing recall information, subsequently referred to as non-recalls.

In these analyses, we considered the following risk factors: multistate exposure outbreaks, supply chain contamination stage, pathogen etiology, and IFSAC Level 1 category food products. We analyzed multistate exposure outbreaks by using a binary variable where single-state exposure outbreaks were the reference. We analyzed supply chain contamination stage by using a 3-level categorical variable (i.e., before preparation, preparation, or unknown) where before preparation was the reference. We analyzed IFSAC Level 1 category food products by using a 4-level categorical variable (i.e., land animals, aquatic animals, plants, or other) where land animals were the reference. We restricted our analyses and independently evaluated 5 etiologies (i.e., *Salmonella*, *E. coli*, *Listeria*, norovirus, and scombroid toxin), as they attributed to 46.8% of all outbreaks and 78.7% of all recalls. We analyzed pathogen etiology by using a binary variable indicating whether the specific pathogen was associated with the outbreak or not.

First, we continued to explore patterns of missingness among risk factors, using frequency tables. Second, we compared differences in frequencies of recalls and non-recalls. Third, we examined the likelihood of a recall with each factor, using univariate logistic regression models. Lastly, we performed multivariate models in a stepwise order, where parameters were specified in accordance with univariate findings:$$\mathrm{Model}\;6:\;\ln\left(Pr\left[R_{i}\right]\right)=\beta_{0}+\beta_{1}\left(S_{i}\right)+\beta_{2}\left(C_{i}\right)+\beta_{3}\left(F_{i}\right)+\beta_{4}\left(D_{i}\right)$$
where *R_i_* is a recall for *i*-outbreak (reference: non-recall and missing combined); *S_i_* is a binary variable indicating multistate exposure of illness for *i*-outbreak; *C_i_* is a categorical variable indicating the supply chain contamination stage of *i*-outbreak; *F_i_* is the IFSAC Level 1 category for *i*-outbreak; and *D_i_* is a binary variable indicating specific etiology associated with *i*-outbreak.

In a sub-analysis, we examined the likelihood of identifying recalls (*n* = 305) during the before-preparation supply chain stage, using logistic regression models and the following risk factors: IFSACL Level 1 category, pathogen etiology, and preparation and consumption locations. We created a dichotomous variable for contamination stage (i.e., before preparation or preparation) by setting outbreaks of unknown preparation stage to missing and using preparation stage as the reference. We restricted our analyses to the 3 most common locations for preparation and consumption (i.e., home, diner, restaurant and other), which accounted for 38.7% and 54.1% of all outbreaks and all recalls, respectively. We analyzed preparation and consumption locations by using a binary variable indicating whether the specific location was associated with the outbreak resulting in a recall or not.

We explored associations between supply chain stage and outbreak etiology, food product, and location of preparation or consumption:$$\mathrm{Model}\;7:\;\ln\left(Pr\left[C_{r}\right]\right)=\beta_{0}+\beta_{1}\left(F_{r}\right)+\beta_{2}\left(D_{r}\right)+\beta_{3}\left(L_{r}\right)$$
where *C_r_* is the *r*-recall identified in before-preparation supply chain stage; *F_r_* is the IFSAC Level 1 category for *i*-outbreak; *D_r_* is a binary variable indicating specific etiology associated with *i*-outbreak; and *L_r_* is a binary variable indicating specific locations where persons prepared or consumed contaminated foods associated with *r*-recall. 

We defined statistical significance as α < 0.05. We evaluated model goodness-of-fit for all models by using the Akaike’s Information Criterion (AIC). We performed data extraction, alignment, management, and cleaning by using Excel 2016 Version 16.59 and Stata SE/16.1 software. We conducted statistical analyses and created data visualizations by using Stata SE/16.1 and RStudio Version 1.2.5042 software.

## 3. Results

### 3.1. Investigating Temporal Trends

NORS reported 22,792 outbreaks from 1 January 1998 to 31 December 2019, of which 305 (1.3%) resulted in food recalls, 13,109 (57.5%) stated no recall, and 9378 (41.1%) had information missing. The initiation of recall reporting resulted in an increase of reported recalls from 0.06 (0.01, 0.23) to 1.23 (0.42, 3.48) in Period 2 and to 1.79 (<0.01, 1566.96) in Period 3 (Table 2).

The estimated monthly mean of outbreaks declined from 106.68 (84.62, 134.52) in Period 1 to 92.53 (78.15, 109.68) in Period 2 (*p* < 0.001) and to 71.05 (34.47, 146.55) in Period 3 (*p* < 0.001). The estimated monthly means of non-recalls were higher in Period 2 (79.12 (54.33, and 115.73)) compared to Periods 1 and 3 (7.49 (5.39, 10.40), *p* < 0.001, and 64.03 (12.75, 103.11), *p* < 0.001, respectively), whereas outbreaks missing recall information were lower in Period 2 (2.18 (0.01, 312.28)) compared to Periods 1 and 3 (99.03 (44.96, 219.89), *p* < 0.001) and 5.12 (0.09, 284.86), *p* = 0.16, respectively) (Table 2 and Figure 1).

Across the entire study period, outbreaks, non-recalls, and outbreaks missing recall information increased by 0.06%/month, 0.24%/month, and 1.90%/month, respectively, whereas recalls decreased by 0.01%/month (Appendix A; Figure 1). We found no significant linear or quadratic trends in monthly outbreaks in Period 1, though outbreaks with missing recall information steadily decreased by 1.03%/month. Though recall information was not formally collected until November 2004, NORS does report non-recalls consistently from January 1998 to November 2004. Non-recalls in Period 1 decreased at an accelerating rate of 1.78%/month (−3.20, −0.34); *p* = 0.014). Unexpectedly, before the official collection of recall status data, in Period 1, NORS reported one recall in April 1998, June 2002, and April 2004; and two recalls in June 2004.

In Period 2, outbreaks increased at a decelerating rate, by 2.13%/month (0.64, 3.65), which continued to increase during Period 3, though at an accelerating rate (2.70%/month (1.46, 3.94)). Similarly, non-recalls increased at a decelerating rate by 10.36%/month (7.76, 13.03) in Period 2, followed by increases at an accelerating rate in Period 3 (9.30%/month (7.64, 10.98)). In contrast, outbreaks with missing recall information decreased across both periods, first at an accelerating rate in Period 2 (25.73%/month (−28.25, −23.20) and then at a decelerating rate in Period 3 (12.65%/month (−15.67, −9.53)). Though we found no significant trends in Period 2, recalls steadily decreased by 3.02%/month (−4.35, −1.76) in Period 3.

Across critical periods, we found that outbreaks decreased by 0.77%/month from Periods 1 to 2 and increased by 0.16%/month and 0.72%/month from Periods 2 to 3 and Periods 1 to 3, respectively. Similarly, outbreaks missing recall information decreased by 6.87%/month from Periods 1 to 2 but increased by 2.97%/month from Periods 2 to 3 and 7.77%/month from Periods 1 to 3. In contrast, both non-recalls and recalls increased by 2.14%/month and 3.02%/month, respectively, between Periods 1 and 2, followed by decreases of similar magnitudes from Periods 2 to 3 (2.14%/month and 2.88%/month, respectively). Both non-recalls and recalls increased slightly between Periods 1 and 3 (0.27%/month and 1.15%/month, respectively).

When examining trend contributions and modeling diagnostics, we found that linear trends contributed to 96.6–98.5% of the overall trend for all models compared to just 1.5–3.4% for quadratic terms (Appendix A). The model fit improved in Model 4, as indicated by a ~0.52–12.4% reduction in AIC for all outcomes. These findings suggested the need for inclusion of quadratic terms when examining seasonal patterns of outcomes.

Outbreaks, non-recalls, and outbreaks with missing recall information demonstrated significant seasonality in at least one critical period (Appendix A; Figure 2). While seasonal patterns of outbreaks appeared visually in all periods, harmonic terms were only significant in Period 1. Non-recalls had significant seasonal patterns in both Periods 1 and 2, whereas outbreaks with missing recall information had significant seasonal patterns in Period 2 only. Though insignificant, outbreaks with missing recalls appeared to have a seasonal pattern in Period 1, also with maximum counts reported in both May and December. All outcomes shared similar patterns, such that maximum counts occurred in April/May, while minimum counts occurred in September/October.

### 3.2. Comparing Risk Factors—Food Recalls

The temporal analyses showed that the reporting of recalls and non-recalls began in Period 2 and continued through Period 3. In comparison, outbreaks missing recall information largely occurred in Period 1, with minimal reporting during Periods 2 and 3. Due to the opposite trends seen in non-recalls and outbreaks missing recall information over the study period, we continued to explore missingness amongst risk factors for outbreaks resulting in a recall with those resulting in non-recalls combined with outbreaks missing recall information.

We found extensive missing data among outbreak risk factors (Table 3). Only 7.6% of outbreaks (51.5% of recalls and 7.0% of non-recalls) had non-missing records for all factors (Figure 3). The location of outbreak exposure had no missing data in our study period. In contrast, 75.9% of outbreaks (*n* = 17,292) had missing supply chain contamination–stage data, including 41.3% of recalls (*n* = 126) and 76.3% of non-recalls (*n* = 17,166). While only 3.28% of recalls had missing etiology information (*n* = 10), nearly one-third of non-recalls failed to report this risk factor (*n* = 7383; 32.4%). Similarly, 12.8% of recalls (*n* = 39) failed to report IFSAC Level 1 information compared to 68.6% of non-recalls (*n* = 15,459). In a sub-analysis of IFSAC, reporting proved even scarcer in further disaggregated subcategories, with 3.08% of outbreaks missing IFSAC Level 2 (*n* = 125 of 4058 outbreaks) and 20.92% of outbreaks missing IFSAC Level 3 (*n* = 823 of 3933 outbreaks) (Figure 4). Both recalls and non-recalls had limited missing records for the preparation (9.51% and 5.01%, respectively) and consumption location (10.16% and 5.23%, respectively) of contaminated foods.

We found that 58.3% of recalls (*n* = 164) were the result of multistate exposure outbreaks compared to only 1.70% of non-recalls (*n* = 382; Table 4). The univariate analyses demonstrated that the odds of multistate exposure outbreaks resulting in a recall were 24.75 times (18.87, 32.55; *p* < 0.001) that of single-state exposure outbreaks—the single-most influential risk factor found. Similarly, 30.16% of recalls (*n* = 92) were associated with plant food products, compared to 5.36% of non-recalls (*n* = 1205); the odds of plant foods resulting in recall were 74% higher (OR = 1.74, 1.31, 2.31; *p* < 0.001) than outbreaks associated with land animals or their byproducts. In contrast, only 2.62% of recalls (*n* = 8) occurred within the preparation supply chain stage, whereas 47.21% of recalls (*n* = 144) occurred within the before-preparation stage. We found that the odds of recall following a preparation-stage outbreak were 95% lower (OR = 0.05, 0.01, 0.11; *p* < 0.001) than a recall following a before-preparation-stage outbreak.

Norovirus and *Salmonella* accounted for 28.74% (*n* = 6550) and 13.04% (*n* = 2972) of all outbreaks. *Salmonella*, *E. coli*, and *Listeria* outbreaks accounted for 33.44% (*n* = 102), 22.62% (*n* = 69), and 9.84% (*n* = 30) of recalls compared to only 12.76% (*n* = 2870), 2.68% (*n* = 603), and 0.30% (*n* = 67) of non-recalls, respectively. The odds of *Salmonella*-, *E. coli*-, and *Listeria*-associated outbreaks resulting in a recall were 1.91 times (1.46, 2.49), 5.27 times (3.86, 7.12), and 16.83 times (9.79, 28.79) that of non-*Salmonella*, non–*E. coli*, and non-*Listeria* outbreaks, respectively (*p* < 0.001). In contrast, norovirus outbreaks accounted for only 8.20% of recalls (*n* = 25) with 57% lower odds (OR = 0.43, 0.26, 0.67) of resulting in a recall compared to non-norovirus outbreaks (*p* < 0.001). The odds of scombroid-poisoning-associated outbreaks resulting in a recall was 46% lower (OR = 0.54, 0.29, 0.90) compared to non-scombroid-poisoning-associated outbreaks.

Locations where contaminated foods were prepared had nearly identical patterns with respect to recall status as with consumption locations. We found that 17.70% (*n* = 54) and 18.69% (*n* = 57) of recalls had contaminated foods prepared or consumed, respectively, in multiple locations compared to only 7.16% (*n* = 1611) and 4.41% (*n* = 991) of non-recalls. Outbreaks with multiple locations for preparation and consumption had odds of 3.59 times (2.57, 4.93) and 4.75 times (3.39, 6.56), respectively, that of single preparation or consumption location outbreaks to result in a recall. Similarly, we found that 22.30% (*n* = 68) and 40.33% (*n* = 123) of recalls were either prepared or consumed at the home, respectively. Outbreaks with at-home preparation and consumption had odds of 1.36 times (1.00, 1.83) and 2.11 times (1.62, 2.73), respectively, that of outbreaks with away-from-home preparation or consumption to result in a recall. In contrast, outbreaks with food preparation or consumption at restaurants had 80–82% lower odds (0.11, 0.32) of resulting in a recall compared to non-restaurant outbreaks.

We found similar patterns when examining the combined effect of all risk factors, with fully adjusted multivariate models having the lowest reported AIC values (Table 5; Appendix A). Irrespective of contaminant etiology, multistate exposure outbreaks had odds that were 11.00–13.50 times (7.00, 21.60) that of single-state exposure outbreaks to result in recall (*p* < 0.001). In contrast, outbreaks where supply chain contamination occurred in the preparation and unknown stages had 93–97% and 53–62% lower odds, respectively, of resulting in a recall (*p* < 0.05) compared to the before-preparation stage. Though outbreaks associated with other foods had significantly greater odds to result in recall compared to land animals, we assumed that the results were spurious due to small sample size within this category.

Across contaminant etiologies, we found that *Listeria*- and norovirus-associated outbreaks had odds of 5.81 times (2.20, 16.40) and 4.93 times (2.39, 9.82) that of non-*Listeria* and non-norovirus outbreaks of resulting in recall, respectively (*p* < 0.001). Though of a lesser magnitude, *E. coli*–associated outbreaks had similarly higher odds of 1.86 (1.08, 3.18) resulting in a recall compared to non–*E. coli* outbreaks. We found no significant findings for either *Salmonella*- or scombroid-poisoning-associated outbreaks.

### 3.3. Comparing Risk Factors—Supply Chain Contamination Stage

After comparing risk factors by recall status, we aimed to examine the likelihood of supply chain contamination in the preparation stage compared to the before-preparation stage among recalls. This analysis would have provided critical information on where within the supply chain recalls commonly occur to inform guidelines for improving outbreak analytics to enhance food traceability in accordance with the 2020 New Era of Smarter Food Safety Blueprint [21,22]. However, due to an insufficient sample size, we were unable to perform these logistic regression analyses.

Of the 305 recalls identified in our study period, 144 recalls (47.21%) were identified in the before-preparation supply chain stage, while only 8 and 27 recalls (5.56% and 8.85%, respectively) were identified in the preparation or unknown stages. Among the before-preparation stage recalls, we found that 20.83% and 28.47% (*n* = 30 and *n* = 41, respectively) occurred within the pre-harvest and pre-processing stages, respectively, compared to 36.73% (*n* = 396) and 9.46% (*n* = 102) of the 1078 before preparation stage non-recalls.

## 4. Discussion

Our study demonstrated the usability of CDC foodborne national surveillance records for investigating food recalls. In doing so, we described temporal trends of recalls for the past two decades and identified risk factors most likely to drive recall occurrence. We found that, while improving since the transition from eFORS to NORS, recall records and information on recall-related risk factors were largely incomplete. Approximately 41.1% (*n* = 9378) of the 22,792 outbreaks reported from 1 January 1998 to 31 December 2019 had a missing recall status. However, our findings suggest that outbreaks missing recall information occurred most frequently before November 2004, with substantial improvements after November 2004 and January 2009, following changes in data-collection methods and reporting standards. Furthermore, only 7.6% of outbreaks (51.5% of recalls and 7.0% of non-recalls) had non-missing records for all factors. These findings alone suggest that current publicly available surveillance records may be insufficient to adequately investigate the financial and human-health burdens of food recalls and foodborne/waterborne outbreaks more broadly.

The New Era of Smarter Food Safety Blueprint aims to enhance interagency communications, design interoperable tools, and improve the timeliness of foodborne outbreak responses [21,22]. While acknowledging the importance of data quality, the Blueprint fails to promote interagency harmonization of existing recall surveillance systems between the FDA, USDA, and CDC. Both the FDA and USDA report traceback information on expenses, manufacturers, and volume of recalled foods not currently traced by the CDC [2,12,13], whereas the CDC traceback investigations identify supply chain contamination locations and where ill persons prepared and consumed contaminated foods. Harmonizing recall record data across these agencies could lead to more comprehensive estimates of healthcare and economic burdens, a better understanding of the impact food recalls has on food waste, and predictive analytics of foodborne outbreaks.

Furthermore, reporting standards impede the ease of temporally or spatially aligning data across agencies or other environmental datasets. In contrast, eFORS and NORS provide comprehensive information on all foodborne outbreaks, thus enabling both descriptions of temporal trends and comparisons of recall-associated risk factors. However, 41.3% and 12.8% of recall-associated records lack information on supply chain contamination stage and IFSAC Level 1 grouping. Other food- and waterborne disease research has explored the supplementation of food-safety surveillance systems with hospitalization records for more precise and complete reporting of notifiable diseases [41,42]. By refocusing collaborative efforts toward interdepartmental data harmonization and considering triangulation of additional public-health-system data, these agencies can create more comprehensive and complete outbreak and recall surveillance vital to understanding food traceability at refined spatiotemporal scales.

Though the volume and velocity of newly reported data increase annually, the CDC must continue to allocate fiscal and personnel resources to check the quality and accuracy of reported data. From January 1998 to November 2004, we found a consistent decrease in the reporting of outbreaks with missing recall information. However, we also found consistent reporting of non-recalls and five reported recalls, despite these records preceding the formal mandate to conduct recall traceback investigations. These findings may reflect the CDC’s attempt to modify historic records, as this will greatly improve the precision and accuracy of temporal trend and risk factor analyses on recalls in future studies. However, these findings may also reflect reporting anomalies requiring further investigation by CDC data-quality and accuracy personnel. Overall, our finding further illustrates the need for interagency collaboration on and greater attention to improving the quality of existing data amidst plans for strengthening surveillance capacity [21,22].

In fact, temporal trends from November 2004 to December 2019 already demonstrated the advantages of regulatory oversight and enforcement of improved data-reporting protocols. After the standardization of reporting of food recalls, found that outbreaks with missing recall information decreased at an accelerating rate, by ~25%/month, while non-recalls decreased at a decelerating rate, by ~10%/month. The expansion of surveillance capacity from eFORS to NORS brought further reductions in outbreaks with missing recall information at a decelerating rate, by ~13%/month. Such extensive reductions in missing recall information over time illustrate the importance of standardized outbreak surveillance reporting and improved usability of CDC surveillance data for investigating food recalls over time.

In our study period, recalls increased by ~3%/month after the beginning of standard traceback investigations and by ~3%/month, again, after the transition to NORS. However, our trend analysis demonstrated that recalls steadily decreased by ~3%/month from January 2009 to December 2019. These trends reflect the improvements in food traceability in the supply chain and, thereby, the mitigating of food recalls after the enactment of the Food Safety Modernization Act in 2009 [43]. Signed into law in 2011, this legislation enabled the FDA to impose mandatory produce safety standards, controls, and inspections for potential hazards in food production, distribution, transport, and retail facilities [44]. Subsequent appendices to the law have mandated increased frequency of food safety inspections, distribution of supply chain records, and testing of food company products to improve early detection and warning of potential outbreaks [45]. Continued support for regulatory oversight and technological advancement on food traceability throughout the supply chain is critical for the continued reduction and prevention of recall events.

Investigations of seasonality can also improve emergency and incident response coordination and enhance early warnings of foodborne outbreaks. In our prior work, we demonstrated stable seasonal patterns of foodborne illnesses in the United States and how these patterns can be examined and understood visually [8,27,46,47]. Across all studies, outbreak peak timing ranged from early July to late August for most enteric infections. More recently, we found that foodborne outbreak severity, measured using an 11-metric index score, similarly peaked in June–September (Simpson et al. (Personal Communication)). In this study, we found that foodborne outbreaks slightly preceded illness and outbreak severity peaks, as the maximum count of outbreaks occurred in April/May, with minimal counts in September/October.

These findings suggest that outbreaks and illness may have synchronized seasonal patterns requiring further investigation to determine the exact lags between seasonal peaks of illnesses, outbreaks, outbreak severity, and recalls. The early onset of outbreaks further emphasizes the need for increased product testing, safety inspections, and toxicological-hazard screenings in April–June annually. However, we cannot discount that these temporal patterns may also reflect changes in annual resources and make the efficient identification of contaminated products through the harmonization of trace-back data even more crucial to food safety [48]. As 47.21% of recalls were associated with the before-preparation supply chain stage, our results suggest that food traceability operations and data reporting must more closely target pre-harvest and preprocessing techniques among producers [21,22]. This will improve data completeness and allow for a closer examination of food traceability, using CDC surveillance data, which we could not perform, due to sample size limitations. Improved monitoring of food safety earlier in the supply chain may reduce both the volume and severity of seasonal outbreaks and illnesses.

Among risk factors, we found that multistate exposure outbreaks consistently had odds of ~10–15 times that of single-state exposure outbreaks to result in a recall. This underscores one of the Blueprint’s main directives of enhanced outbreak responsiveness, rapid traceback deployment, and strengthened root-cause analyses to identify the location of outbreaks and recalls [21,22]. In doing so, multistate exposures outbreaks can be more thoroughly contained to minimize the volume and severity of ill persons per recall. Furthermore, improved traceback investigations will better identify food distribution and retail pathways to mitigate the expansiveness of outbreaks within the supply chain. These efforts must more readily target outbreaks associated with *E. coli*, *Listeria*, and norovirus, as these thee etiologies had odds of ~1.5–6 times that of non–*E. coli*, non-*Listeria*, and non-norovirus outbreaks to result in food recalls.

Our study was subject to several limitations. First, recall records were vulnerable to reporting bias of high-priority pathogens and food products that are most burdensome because they cause pathogen-related deaths. In 2015, just five pathogens (i.e., *Salmonella*, *Toxoplasma gondii*, *Listeria monocytogenes*, *Campylobacter*, and norovirus) caused 90% of the economic burden imposed by foodborne outbreaks [49]. Similarly, in this study, we found that *Salmonella*, *Listeria* and norovirus outbreaks predominantly resulted in a food recall. Second, further risk factor analysis by IFSAC Levels 2–5 and the sub-analysis of recalls in the preparation stage compared to the before-preparation stage were not possible, due to insufficient sample size. These analyses would have determined major food types or subtypes and supply chain contamination locations with higher probabilities of contamination resulting in a recall, thus informing the prioritization of traceback food products and locations by regulatory agencies. Next, the pathogen etiology and the location of preparation and consumption of contaminated foods were not originally mutually exclusive variables. By creating multi-pathogen and multi-location food preparation or consumption variables, we might have introduced potential multiplicity when comparing specific pathogen etiology, or location of preparation or consumption to their respective reference groups. Lastly, we paid sufficient attention to missing data and the structure of the missing data [50]. On the surface, we could handle missing data by using imputation; however, due to structural missingness, this could create bias.

To our knowledge, this study is the first to utilize NORS recall record data to examine temporal trends, particularly before and after November 2004 and January 2009. Additionally, our comparison of risk factors between recalls and non-recalls highlighted existing biases in reporting influenced by available resources or outbreak healthcare and economic burden. Future directions should include more granular analyses of contaminant etiology and preparation and consumption locations and explore the relationship of these risk factors on foodborne outbreak severity. With the FDA taking a new approach to food safety via the New Era of Smarter Food Safety Blueprint, we urge food-safety and public-health agencies to collaborate more closely and standardize data-reporting protocols, thereby improving the spatiotemporal alignment and harmonization of publicly reported national surveillance databases on food recalls.

## 5. Conclusions

Food recalls impose an extensive fiscal burden on the food economy in the United States, in addition to recall- and outbreak-associated foodborne illnesses. However, current national surveillance systems lack sufficient data quality and completeness for establishing precise and accurate early outbreak and recall detection and warnings. While data quality has improved over time, as the result of federal food-safety policies, further regulatory oversight is still needed. Future policy regulations must standardize timely and thorough data reporting of food recall and outbreak events to improve the traceability of food throughout the supply chain and responsiveness to multistate exposure outbreak events.

## Figures and Tables

**Figure 1 ijerph-19-04947-f001:**
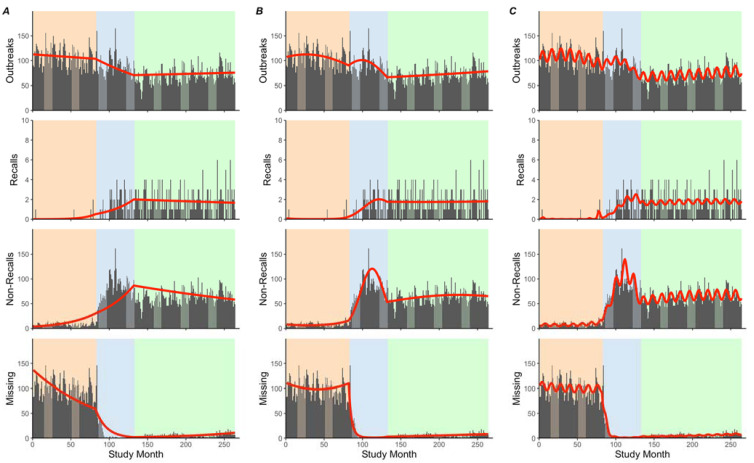
Stacked time-series plots of monthly outbreaks, recalls, non-recalls, and outbreaks with missing recall information (top to bottom rows, respectively), as reported by the electronic Foodborne Outbreak Reporting System (eFORS) and National Outbreak Reporting System (NORS) in 1998–2019. We present segmented time-series model results, adjusting for linear trends only (Model 3; Panel (**A**)); linear and quadratic trends (Model 4; Panel (**B**)); and linear, quadratic, and harmonic trends (Model 5; Panel (**C**)). Within each plot, we report observed counts (gray bars) with fitted model results (red lines) and indicate critical periods by using different background colors.

**Figure 2 ijerph-19-04947-f002:**
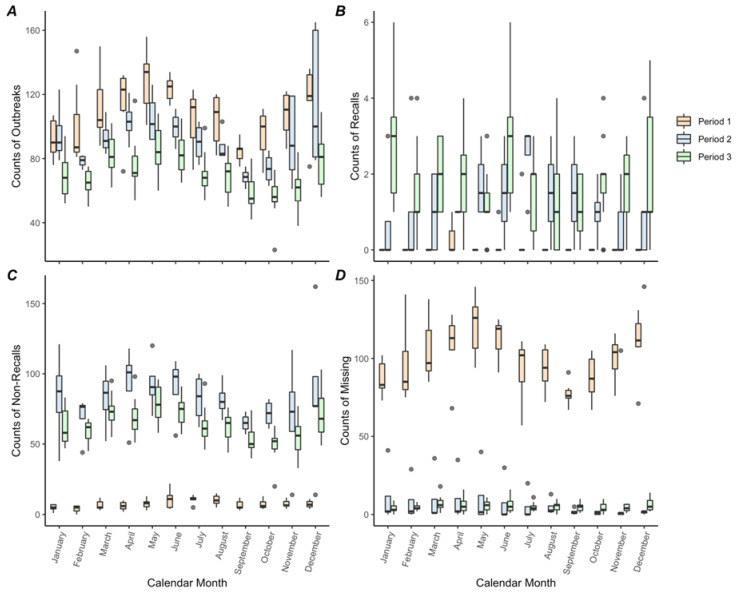
Boxplots of monthly counts of outbreaks (**A**), recalls (**B**), non-recalls (**C**), and outbreaks missing recall information (**D**), as reported by the electronic Foodborne Outbreak Reporting System (eFORS) and National Outbreak Reporting System (NORS) in 1998–2019. Each of the 3 critical periods is indicated by using a different box color. Grey dots represent outliers.

**Figure 3 ijerph-19-04947-f003:**
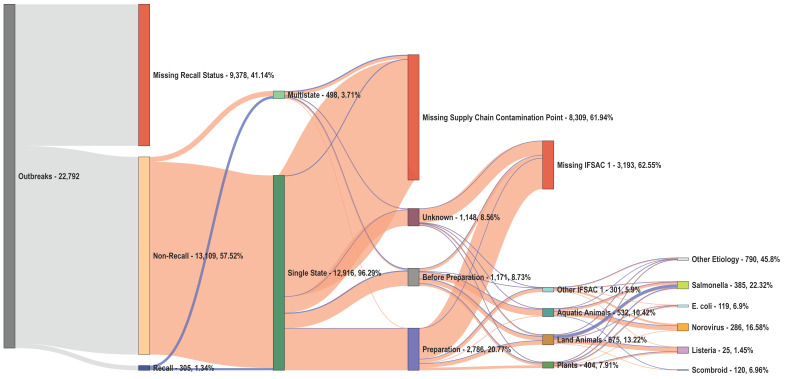
A Sankey Diagram of the distribution of recalls and non-recalls with and without missing information across all risk factors examined by using Model 6. We report frequencies and percentages of foodborne and waterborne outbreaks associated with each risk factor for all 22,792 outbreaks reported by the electronic Foodborne Outbreak Reporting System (eFORS) and National Outbreak Reporting System (NORS) in 1998–2019. Blue and orange colors define recalls and non-recalls, respectively. Outbreaks with missing recall status or risk-factor information are defined with orange terminal nodes. We calculated percentages according to the frequency of observations available for each risk factor, which include recall status, single- or multistate exposure outbreak, supply chain contamination stage, Interagency Food Safety Analytics Collaboration (IFSAC) Level 1 food categorization, and etiology of contaminant. Other IFSAC 1 includes outbreaks associated with Other (*n* = 36), Unclassifiable (*n* = 33), Undetermined (*n* = 229), and Invalid (*n* = 3) food products. For contaminant etiology, we list the 5 etiologies of interest in our study (*Salmonella*, *E. coli*, norovirus, *Listeria*, and scombroid poisoning), as well as Other Etiology to account for contaminants not considered in our analyses.

**Figure 4 ijerph-19-04947-f004:**
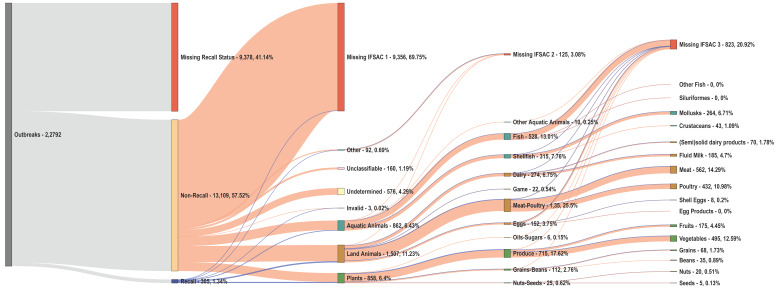
A Sankey Diagram of a sub-analysis examining the distribution of recalls with and without missing information across the Interagency Food Safety Analytics Collaboration (IFSAC) food categorization for Levels 1–3. We report frequencies and percentages of foodborne and waterborne outbreaks associated with each category for all 22,792 outbreaks reported by the electronic Foodborne Outbreak Reporting System (eFORS) and National Outbreak Reporting System (NORS) in 1998–2019. Blue and orange colors define outbreaks with non-missing and missing recall status information, respectively. We calculated percentages according to the frequency of observations in each level. Level 1 describes overarching food groups, including aquatic animals, land animals, plants, and other foodstuffs. Level 2 further categorizes groups into fish/shellfish, other aquatic animals, dairy, game, meat/poultry, eggs, oils/sugars, produce, grains/beans, and seeds/nuts. Level 3 provides more refined categories by specific food subtypes.

**Table 1 ijerph-19-04947-t001:** Critical periods used to examine trends and seasonality of CDC foodborne and waterborne outbreak surveillance records with segmented regression analyses.

Period	Start Date	Duration	Description
1	January 1998	82 Months	Surveillance reporting begins under eFORS.
2	November 2004	50 Months	The CDC’s Investigation of a Foodborne Outbreak revised to include if any food product was recalled from an outbreak.
3	January 2009	132 Months	The CDC transitions reporting foodborne outbreak data from eFORS to NORS.

**Table 2 ijerph-19-04947-t002:** Summary statistics for monthly outbreaks, recalls, non-recalls, and outbreaks missing recall information, as reported by the electronic Foodborne Outbreak Reporting System (eFORS) and National Outbreak Reporting System (NORS) in 1998–2019.

Statistic	Full Study Period January 1998–December 2019	Period 1 January 1998–October 2004	Period 2 November 2004–December 2008	Period 3 January 2009–December 2019
Outbreaks				
Mean (95% CI)	86.33 (83.45, 89.33)	106.68 (84.62, 134.52)	92.53 (78.15, 109.68)	71.05 (34.47, 146.55)
Median (LQR, UQR)	82.00 (68.00, 103.25)	109.00 (87.00, 121.00)	87.00 (79.00, 103.00)	69.00 (60.00, 81.00)
Min, Max	23, 165	71, 156	61, 165	23, 116
L-Skew, L-Kurt	0.12, 0.11	0.02, 0.04	0.22, 0.19	0.09, 0.13
Recalls				
Mean (95% CI)	1.15 (1.00, 1.34)	0.06 (0.01, 0.23)	1.23 (0.42, 3.48)	1.79 (<0.01, 1566.96)
Median (LQR, UQR)	1.00 (0.00, 2.00)	0.00 (0.00, 0.00)	1.00 (0.00, 2.00)	2.00 (1.00, 3.00)
Min, Max	0, 6	0, 2	0, 4	0, 6
L-Skew, L-Kurt	0.30, 0.02	0.94, 0.85	0.20, <0.01	0.13, 0.11
Non-Recalls				
Mean (95% CI)	49.65 (44.69, 55.18)	7.49 (5.39, 10.40)	79.12 (54.33, 115.73)	64.03 (12.75, 103.11)
Median (LQR, UQR)	57.00 (11.00, 73.00)	6.50 (5.00, 10.00)	78.50 (66.25, 97.25)	63.00 (54.00, 74.00)
Min, Max	0, 162	0, 22	14, 162	20, 103
L-Skew, L-Kurt	−0.01, 0.01	0.14, 0.09	-0.009, 0.20	0.07, 0.13
Outbreaks Missing Recall Information				
Mean (95% CI)	35.52 (29.62, 42.60)	99.03 (44.96, 219.89)	2.18 (0.01, 312.28)	5.12 (0.09, 284.86)
Median (LQR, UQR)	7.00 (3.00, 81.00)	101.00 (83.25, 112.50)	1.00 (0.00, 3.00)	4.00 (3.00, 18.00)
Min, Max	0, 146	57, 146	0, 146	0, 18
L-Skew, L-Kurt	0.39, 0.01	0.03, 0.07	0.75, 0.52	0.20, 0.13

95% CI, 95% confidence interval; LQR, lower quartile range; UQR, upper quartile range; min, minimum; max, maximum; L-Skew, L-skewness; L-Kurt, L-kurtosis.

**Table 3 ijerph-19-04947-t003:** Frequency and percentage of outbreaks with missing information by recall status (total, recall, and non-recall) and risk factor. We extracted data from the electronic Foodborne Outbreak Reporting System (eFORS) and National Outbreak Reporting System (NORS) in 1998–2019. Risk factors include location of residence for ill persons, etiology of contaminant, location of preparation and consumption of contaminated foods, Interagency Food Safety Analytics Collaboration (IFSAC) Level 1 food categorization, and supply chain contamination stage. We list risk factors in ascending order by percentage of outbreaks with missing information.

Outbreak	Recall (*n* = 305)		Non-Recall (*n* = 22,487)		Total (*n* = 22,792)	
Risk Factors	*n*	%	*n*	%	*n*	%
Location of Outbreak Exposure	0	0.00	0	0.00	0	0.00
Etiology	10	3.28	7373	32.79	7383	32.39
Preparation Location	29	9.51	1127	5.01	1156	5.07
Consumption Location	31	10.16	1175	5.23	1206	5.29
IFSAC Level 1	39	12.79	15,420	68.57	15,459	67.83
Supply Chain Contamination Stage	126	41.31	17,166	76.34	17,292	75.87

**Table 4 ijerph-19-04947-t004:** Frequency and percentage of foodborne and waterborne outbreaks, overall, by recall status (recall vs. non-recall) and by outbreak risk factors. We extracted data from the electronic Foodborne Outbreak Reporting System (eFORS) and National Outbreak Reporting System (NORS) in 1998–2019. Risk factors include location of outbreak exposure, supply chain contamination stage, Interagency Food Safety Analytics Collaboration (IFSAC) Level 1 food categorization, contaminant etiology, and locations of preparation and consumption of contaminated foods. We supplement descriptive statistics with odds ratio estimates (and 95% confidence intervals) from univariate logistic regressions.

Outbreak	Recall (*n* = 305)		Non-Recall (*n* = 22,487)		Total (*n* = 22,792)		Univariate Odds Ratio
Risk Factors	*n*	%	*n*	%	*n*	%	95% Conf. Int.
Location of Outbreak Exposure							
Single-State							1
Multistate	164	53.77	382	1.70	546	2.40	24.75 (18.87, 32.55) ^a^
Supply Chain Contamination Stage							
Before Preparation	144	47.21	1078	4.79	1222	5.36	1
Preparation	8	2.62	3022	13.44	3030	13.29	0.05 (0.01, 0.11) ^a^
Unknown	27	8.85	1221	5.43	1248	5.48	0.88 (0.54, 1.39)
IFSAC Level 1							
Land Animals	111	36.39	2534	11.27	2645	11.60	1
Aquatic Animals	56	18.36	1506	6.70	1562	6.85	0.85 (0.61, 1.17)
Plants	92	30.16	1205	5.36	1297	5.69	1.74 (1.31, 2.31) ^a^
Other Foods	6	1.97	181	0.80	187	0.82	0.76 (0.29, 1.60)
Contaminant Etiology							
Non-*Salmonella*	193	63.28	12,244	54.45	12,437	54.57	1
*Salmonella*	102	33.44	2870	12.76	2972	13.04	1.91 (1.46, 2.49) ^a^
Non–*E. coli*	226	74.10	14,511	64.53	14,737	64.66	1
*E. coli*	69	22.62	603	2.68	672	2.95	5.27 (3.86, 7.12) ^a^
Non-*Listeria*	265	86.89	15,047	66.91	15,312	67.18	1
*Listeria*	30	9.84	67	0.30	97	0.43	16.83 (9.79, 28.79) ^a^
Non-Norovirus	270	88.52	8589	38.20	8859	38.87	1
Norovirus	25	8.20	6525	29.02	6550	28.74	0.43 (0.26, 0.67) ^a^
Non-Scombroid Poisoning	281	92.13	14,668	65.23	14,949	65.59	1
Scombroid Poisoning	14	4.59	446	1.98	460	2.02	0.54 (0.29, 0.90) ^b^
Preparation Location							
Non-Home	208	68.20	19,460	86.54	19,668	86.29	1
Home	68	22.30	1900	8.45	1968	8.63	1.36 (1.00, 1.83) ^b^
Non-Diner	247	80.98	17,597	78.25	17,844	78.29	1
Diner	29	9.51	3763	16.73	3792	16.64	0.78 (0.51, 1.17)
Non-Restaurant	255	83.61	13,128	58.38	13,383	58.72	1
Restaurant	21	6.89	8232	36.61	8253	36.21	0.18 (0.11, 0.27) ^a^
Single Location	222	72.79	19,749	87.82	19,971	87.62	1
Multiple Locations	54	17.70	1611	7.16	1665	7.31	3.59 (2.57, 4.93)
Consumption Location							
Non-Home	151	49.51	17,876	79.49	18,027	79.09	1
Home	123	40.33	3436	15.28	3559	15.62	2.11 (1.62, 2.73) ^a^
Non-Diner	249	81.64	17,872	79.48	18,121	79.51	1
Diner	25	8.20	3440	15.30	3465	15.20	0.77 (0.48, 1.16)
Non-Restaurant	257	84.26	14,863	66.10	15,120	66.34	1
Restaurant	17	5.57	6449	28.68	6466	28.37	0.20 (0.11, 0.32) ^a^
Single Location	217	71.15	20,321	90.37	20,538	90.11	1
Multiple Locations	57	18.69	991	4.41	1048	4.60	4.75 (3.39, 6.56) ^a^

Superscripts indicate statistical significance at *p* < 0.001 (^a^), and *p* < 0.05 (^b^).

**Table 5 ijerph-19-04947-t005:** Logistic regression results examining the likelihood of foodborne and waterborne outbreaks resulting in food recalls, as reported by the electronic Foodborne Outbreak Reporting System (eFORS) and National Outbreak Reporting System (NORS) in 1998–2019. We selected risk factors according to univariate logistic regression results and added factors in a stepwise order. Risk factors include multistate exposure outbreaks (reference: single-state exposure outbreaks), supply chain contamination stage (reference: before-preparation stage), IFSAC Level 1 food categories (reference: land animals), and presence of a contaminant etiology (reference: absence of or unknown etiology). We report fully specified models for 5 contaminant etiologies, namely *Salmonella*, *E. coli*, *Listeria*, norovirus, and scombroid poisoning associated outbreaks. We report the odds ratio estimates (and 95% confidence intervals), Akaike’s Information Criterion (AIC), and the number of observations per model.

Risk Factors	*Salmonella*	*E. coli*	*Listeria*	Norovirus	Scombroid Poisoning
Location of Outbreak Exposure					
Multistate	13.00 (8.17, 21.00) ^a^	11.60 (7.39, 18.50) ^a^	11.00 (7.00, 17.60) ^a^	13.50 (8.59, 21.60) ^a^	12.60 (8.05, 20.10) ^a^
Supply Chain Contamination Stage					
Preparation	0.06 (0.02, 0.17) ^a^	0.07 (0.02, 0.19) ^a^	0.06 (0.02, 0.16) ^a^	0.03 (0.01, 0.10) ^a^	0.06 (0.02, 0.17) ^a^
Unknown	0.44 (0.25, 0.77) ^b^	0.47 (0.26, 0.82) ^b^	0.38 (0.20, 0.68) ^b^	0.41 (0.23, 0.72) ^b^	0.43 (0.24, 0.75) ^b^
IFSAC Level 1					
Aquatic Animals	0.58 (0.35, 0.95) ^b^	0.69 (0.42, 1.13)	0.64 (0.40, 1.03)	0.46 (0.27, 0.77) ^b^	0.54 (0.32, 0.90) ^b^
Plants	0.71 (0.43, 1.14)	0.72 (0.44, 1.16)	0.72 (0.43, 1.17)	0.67 (0.41, 1.08)	0.71 (0.43, 1.15)
Other Foods	6.45 (1.17, 29.2) ^b^	6.94 (1.26, 31.50) ^b^	7.16 (1.31, 32.60) ^b^	4.91 (0.90, 22.30) ^b^	6.53 (1.18, 29.70) ^b^
Etiology					
Etiology Present	0.84 (0.53, 1.31)	1.86 (1.08, 3.18) ^b^	5.81 (2.20, 16.40) ^a^	4.93 (2.39, 9.82) ^a^	1.76 (0.77, 3.73)
Modeling Diagnostics					
AIC	782.78	778.41	770.48	766.33	781.52
Observations	1596	1596	1596	1596	1596

Superscripts indicate statistical significance at *p* < 0.001 (^a^), and *p* < 0.05 (^b^).

## Data Availability

The Centers for Disease Control and Prevention (CDC) publicly report records for the electronic Foodborne Outbreak Reporting System (eFORS) and National Outbreak Reporting System (NORS) on their data dashboard [36]. We received more detailed records than provided publicly through a formal data request with the NORS Foodborne and Animal Contact Team.

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
