# Peer review of "Exploring Risk Factors of Recall-Associated Foodborne Disease Outbreaks in the United States, 2009–2019"

_ijerph, 2022, doi:10.3390/ijerph19094947_

Round 1
Reviewer 1 Report
This is an interesting manuscript that illustrated the usability of the electronic Foodborne Outbreak Reporting System (eFORS) and National Outbreak Reporting System (NORS) for explaining temporal trends and outbreak risk factors of food recalls in 1998-2019. Authors also explained the limitations of other data sources (e.g. FSIS, FDA) and provide justification to use eFORE and NORS in the context of this study. The core weakness of the manuscript is the theoretical aspect. Authors did not pay much attention in exploring the literature on food product recall in the introduction section and did not write anything as literature review. I would strongly recommend authors to address the following four points in their revised manuscript.
-
- Author did not discuss about what has been done in the existing literature on food product recall and what is the gap in the literature that authors have attempted to address in this manuscript.
- Please explain the key contribution of the manuscript. I could not find anything in the introduction section.
- Authors should write a literature review section on food product recall.
- Please provide a summary table of literature that will not only identify the gap in the literature but also strengthen the contribution of the manuscript.
- Authors did an excellent job in presenting the methodology and result sections which are the key strength of the manuscript.
Author Response
Please see the attached file, specifically the response to Reviewer #1 (pages 1-2).

Reviewer 2 Report
Generally, this is a well-written manuscript. In the text, the authors may be quoting the etiological agents defined by CDC but they do not appear to be familiar with them as they italicize viruses and use simply Escherichia as a pathogen.
Specifically, check whether Adenovirus, Astrovirus, Norovirus and Rotavirus should be italicized and UC letter. They should be written adenovirus, astrovirus, norovirus and rotavirus unless it is the specific Latin name. Use what CDC uses.
Also, Escherichia as a pathogen does not make sense. E. coli is a common commensal and there are many varieties of pathogenic E. coli. Specific pathogen groups need to be shown.
Author Response
Please see the attached file, specifically the response to Reviewer #2 (page 2).

Reviewer 3 Report
Overall Comments
In this paper the authors aim to explore risk factors of associated foodborne outbreaks in the US using data from the CDC NORS and eFORS datasets. Overall the authors completed a significant amount of work in trying to understand these associations, but there are some disjointed thoughts in connecting the value of this work and there are a few major gaps which the authors need to address.
Additional comments I have are to 1) pair down the results (remove some of the analyses and/or don't say literally everything, especially when the information is presented in tables or figures); 2) reformat figures 3 & 4 in a way that makes a static figure easier to read (Sankey Diagrams are meant to be interactive - as it is in the paper, it is very difficult to read and is too busy to interpret); 3) consider recategorizing outbreak risk factors to reduce possibility of multiplicity; and 4) restructure the discussion to focus less on missing data and include strengths/limitations. There is a lot of information and analyses in this paper and there is a disjoint between what results are presented and what is discussed and could be applicable in the field of foodborne epidemiology. This really could be 2-3 papers.
A major revision should take place to focus on more actionable data and reduce redundancy throughout the text.
Abstract
No mention of economic burden in actual analysis - consider reframing 1st sentence to discuss health impact (lines 12-13)
Introduction
Overall
The introduction is informative and well-written. However, beginning the introduction with economic burden may make the reader assume this paper is about economic burden, not temporal trends and risk factors of recall-associated outbreaks. Consider removing or restructuring later on in the introduction. I also think enteric epidemiological data should be included in the introduction, which is more closely aligned with the paper than economic burden.
Line by Line
Line 51 - characteristics of ill consumers comes from CDC, not sure if this should be added or not
Lines 59 – 66: I believe the use of “recalls’” is not appropriate here. I think you could just use recall, or recall record. e.g. “recall record temporal trends and risk factors”
Line 68-69: this is inaccurate, the agencies do report state, and smaller jurisdiction such as city, when appropriate and available. Which spatial scale would provide more meaningful output in order to make your argument more clear? I would argue that their annual and descriptive reports are not meant for time series analyses, but you are able to do these analyses if you get the raw data. Is the issue then that the data is not readily available? Later in your results you show that location data is the least missing (0%) variable.
Lines 77-87 - source for CDC info?
Lines 88-95: There have been many manuscripts published using NORS data for foodborne outbreaks. Here are a few:
This one has looked at waterborne outbreaks: https://ngwa.onlinelibrary.wiley.com/doi/abs/10.1111/gwat.12121
Outbreaks in leafy greens: https://pubmed.ncbi.nlm.nih.gov/25697407/
CDC outbreak reporting summaries: https://pubmed.ncbi.nlm.nih.gov/23804024/
risk factors for STEC outbreaks: https://pubmed.ncbi.nlm.nih.gov/27526280/
Prior NORS data: https://pubmed.ncbi.nlm.nih.gov/15508656/
In terms of seasonality, this FoodNet article is also relevant: https://www.ncbi.nlm.nih.gov/pmc/articles/PMC7562704/
Lines 99-106: These read a lot like Methods and Results rather than an Introduction. You may want to consider moving some of the more detailed information to the Methods section.
Methods
Overall
The methods section was very detailed and informative. Some sentences may be redundant or unnecessary.
Line by Line
114-115: “We received no data…” unnecessary detail
121-123 - remove sentence on duplicates
135,142-143 - redundant?
Table 1 Title: Remove ‘Descriptive Table of the’
181: I am assuming a=start of period 2 and b is 82+50 months to account for the start of period 3? Or is period 3 b It’s unfortunate it worked out that periods 1+2=3. It may be worth clarifying in the text.
Line 184: Model 3 could be better labeled (I would just add ‘Model’ in front of all the right justified numbers. You also use the term ‘Equation’ later in the paragraph. Just be consistent.
Line 212: ‘very few non-recall outbreaks’ may clarify this sentence or move the last part of 217 up where you define ‘non-recalls’
210-214 - move to results section?
218: Why would the presence of a pathogen’s etiology be considered a risk factor for a recall? On a food safety level, not all health departments have the resources to collect swabs or to run extensive testing in order to determine the pathogen. For all of your “risk factors” I think what you’re really getting at is resource allocation, issues with data availability, and onboarding of states into eFORS and NORS, which happened not all at once but on a state-by-state basis. I think these are appropriate, but the motivation for using these isn’t well described in the Introduction. There needs to be a more clear motivation for these methods of understanding missing data in the introduction.
223 - specify stepwise selection was used
226 - what does ‘not clear’ refer to?
Later in line 234 you discuss that you restricted your analyses to known etiologies, so what “presence” are you then estimating?
232, 245 - why use absence or unknown etiology as reference? (more of an internal question)
253-255 - small, but use consistent formats for software versions? (excel version 2103 vs RStudio [1.2.5042])
Results
Overall
Very detailed reporting of results with useful information. A couple of general observations: there are some inconsistencies in results reporting formats; the categorization of some risk factors is a little confusing (e.g., prep and consumption categories, pathogen etiology - increased potential for multiplicity?). The authors also present a lot of information on missing data, which I understand is important to elucidate issues to hopefully create change, but the overall findings may have been more robust with an additional analysis using imputation.
Line by Line
260-262: Gradual increase of reported recalls – is this due to reporting requirements increasing for states, or because of a true increase in recalls? I comment on this further in the discussion section below.
This is small, but there are inconsistencies in reporting results (e.g., line 268 uses double parentheses while line 281 uses parentheses and brackets)
316-319 - methods?
322 - least missing data (0.00%, inconsistent sig figs) → no missing data?
341-344 - isn’t the preparation stage reported (n=144) the before preparation stage (wording is a little confusing)
345-384: Since you are using a logistic regression with odds ratios, you should interpret the ORs in terms of odds, not using “times as likely” which is more appropriate for a relative risk.
345-354 - may be more informative and easier on the reader to explicitly state the ref categories
381 - specify ref categories to make it easier on the reader?
Tables and Figures
Table 2: Remove ‘Descriptive statistics of’. Delete sentence ‘We defined…” this information is already in the text. Definitions of LQR and UQR, etc could be table footnotes instead of in the title.
Table 2 - median of 0.00 for period 1?; inconsistent sig figs
Table 2: Is ‘Outbreaks’ all outbreaks or just those that triggered a recall? If so, it may be useful to use the label ‘Outbreaks with Recalls’
Figure 1: Delete ‘Multi-panel, shared axis’. Unless the journal requires it, there could be far less info in the legend text.
Figure 1: Nice figure overall!
Table 4 - preparation and consumption location categories are a little confusing, I know the authors mentioned categories are not mutually exclusive but would it not make more sense to compare diner to restaurant to home? Same with contaminant etiology
Table 5 - specify that ORs are reported in the table itself, not just description?
Discussion
Overall
The conclusion was also well-written and included many call-to-actions. The authors could have included more strengths and limitations of the study (other than missing data) and analyses specifically. Also, there was no mention of economic burden, making the first paragraph of the introduction feel even more disjointed. If the authors keep that information in the introduction, a brief discussion on how earlier identification of outbreaks and reduced recalls can reduce the economic burden would make that point more well-rounded.
Line 400: NORS data is not publicly available, you need to request data for the level of analyses you are doing. Also remove the use of waterborne, as you did not use data from the waterborne outbreaks in your analyses.
Line 405-407: It’s inappropriate to aggregate the percent missing here, as you found a significant difference between your 3 phases in your results. The majority of the missingness, as you described, comes from phase 1, with a significant level of non-missingness in later phases due to the transition. Consider rewriting these results in terms of phases. It is promising that the level of missingness decreased over time with the introduction of electronic reporting systems. You discuss this later, but this statement is misleading.
Lines 411: The use of this tool seems to be an important part of your discussion, but it’s the first we’re hearing of it. I think an overview of what the Blueprint is and why it’s important for food safety, and the work you’re doing here, would be important to include in your introduction.
Line 416: Why would it be important for the CDC to track this information if FDA and USDA track it? It is not in their jurisdiction to do so, but to work with these agencies during outbreak and recall periods. Consider re-writing this sentence in terms of connecting datasets, rather than reporting.
Line 449: investing to investigating
Lines 464-472: An alternate explanation for the count of outbreak peak could be that health departments have the resources early in FBD season to investigate and conduct surveillance for outbreaks. Once May hits, the majority of FBDs increase substantially, and many health departments switch to a percentage of interviewing cases, or only attempt interviewing a portion because they do not have the capacity to conduct surveillance for all of them. If they aren’t doing surveillance, they aren’t finding outbreaks and they aren’t reported. I think this is a reporting issue, not an “early onset of outbreaks.”
Conclusions
Lines 504-506: While I agree with you, I don’t believe that the research reported here supports this conclusion. You did not explore data quality across multiple datasets, or discuss standardized data reporting protocols.
Author Response
Please see the attached file, specifically the response to Reviewer #3 (pages 2-15).

Round 2
Reviewer 3 Report
Thank you for all the time spent on the revisions. I suggest working with the editors to make the Figure showing all the sources and pathogens easier to read in static form.
Nice job.